# Detecting changes in generation and serial intervals under varying pathogen biology, contact patterns and outbreak response

**Rachael Pung**[1,2]*, **Timothy W. Russell**[2], **Adam J. Kucharski**[2]

**1** Ministry of Health, Singapore, Singapore, **2** Centre for the Mathematical Modelling of Infectious Diseases, London School of Hygiene and Tropical Medicine, London, United Kingdom

* rachael.pung@lshtm.ac.uk

**Data Availability Statement:** All relevant data are within the manuscript, its Supporting information files, or in a public repository https://github.com/rachaelpung/generation_intervals.

## Abstract

The epidemiological characteristics of SARS-CoV-2 transmission have changed over the pandemic due to emergence of new variants. A decrease in the generation or serial intervals would imply a shortened transmission timescale and, hence, outbreak response measures would need to expand at a faster rate. However, there are challenges in measuring these intervals. Alongside epidemiological changes, factors like varying delays in outbreak response, social contact patterns, dependence on the growth phase of an outbreak, and effects of exposure to multiple infectors can also influence measured generation or serial intervals. To guide real-time interpretation of variant data, we simulated concurrent changes in the aforementioned factors and estimated the statistical power to detect a change in the generation and serial interval. We compared our findings to the reported decrease or lack thereof in the generation and serial intervals of different SARS-CoV-2 variants. Our study helps to clarify contradictory outbreak observations and informs the required sample sizes under certain outbreak conditions to ensure that future studies of generation and serial intervals are adequately powered.

## Author summary

Generation and serial intervals quantify the timescale of a transmission process from one person to another. In turn, this informs the speed required to expand outbreak control efforts, especially when we encounter a change in the biological properties of the pathogen. However, shifts in human contact patterns and evolving outbreak response measures can collectively bias the interpretation of these intervals. Using a simulation framework, we estimated the power to detect a difference in these intervals under the influence of multiple factors and investigated the potential for bias in generation and serial interval estimates for COVID-19.

**Funding:** RP acknowledges funding from the Singapore Ministry of Health. AJK was supported by a Sir Henry Dale Fellowship jointly funded by the Wellcome Trust and the Royal Society (grant 206250/Z/17/Z). The funders had no role in study design, data collection and interpretation, or the decision to submit the work for publication.

**Competing interests:** The authors have declared that no competing interests exist.

## Introduction

When novel SARS-CoV-2 variants of concern have been identified, a crucial question has been how the epidemiology of the emerging variant relates to the current dominant variants. Novel variants may exhibit multiple phenotypic changes, including changes in the viral load trajectory [1–3], incubation period [4,5], generation interval [5,6] and serial interval [7,8]. Quantifying these epidemiological characteristics is essential to interpret the relative transmissibility of variants of concern and, thus, the potential effectiveness of individual and population-level outbreak control measures. However, comparing specific variants can be challenging owing to changing population-level epidemic dynamics, shifts in human contact patterns and evolving outbreak response measures. In turn, these factors can bias conclusions about the extent to which observed changes in variant dynamics are the result of inherent viral properties, rather than characteristics of the population in which they are spreading. Despite efforts to compare different aspects of SARS-CoV-2 variant epidemiology, the potential magnitude and direction of such biases in general remain unclear.

Two epidemiological parameters that are particularly important for interpreting growth patterns are the generation and serial intervals. When variant prevalence grows rapidly within a population, it may be the result of increased transmissibility, a shorter delay from one infection to the next, or both. The generation interval is commonly used to define this transmission timescale (i.e. the average time between infection of infector and infection of infectee). This interval is a combination of both a host's infectiousness profile since time of infection as well as the timing of social contacts between this primary case and potential infectees. However, because infection times are rarely observed, serial intervals (i.e. time between symptoms onset in an infector and an infectee) are often used either as proxies, or to infer the times of infection–and hence the generation interval–over a range of exposure times [9,10]. This can result in several potential biases. Observed serial intervals based on the onset times of infectees are shorter during the exponential phase of an outbreak because transmission events involving infectees with longer incubation periods have yet to be observed [11]. Furthermore, shorter delays from symptom onset-to-isolation of cases over the course of an outbreak truncates the infectiousness profile and, hence, the serial interval [12]. Large-scale movement restrictions could also influence the relative contribution of household and non-household interactions to transmission and, hence, the overall distribution of generation and serial intervals [6,12]. Even if analyses were confined to household contacts only, the timing of contact may not be consistent across the days. As such, broad assumptions such as constant contact over time could potentially diminish ability to detect differences in the generation and serial intervals between existing and novel variants and, hence, distinguish between a more transmissible variant, and merely a faster one [7].

Using a high-resolution dataset on pre-pandemic human social interactions collected from a large-scale UK study of 469 community participants [13], we parameterised a transmission model of SARS-CoV-2 and other epidemic-prone pathogens to understand factors influencing observed differences in generation and serial intervals during outbreaks. We explored factors including varying viral epidemiological characteristics, isolation strategies, epidemic dynamics, contact patterns between pairs of individuals and within household settings (i.e. competing infectors). Furthermore, we estimated the statistical power to detect these differences between variants and, hence, the potential for bias in variant estimates, using the COVID-19 pandemic as a case study.

## Results

To understand changes in estimated generation and serial intervals, we simulated the incubation period of infector and infectee pairs, and modelled stochastic transmission based on the

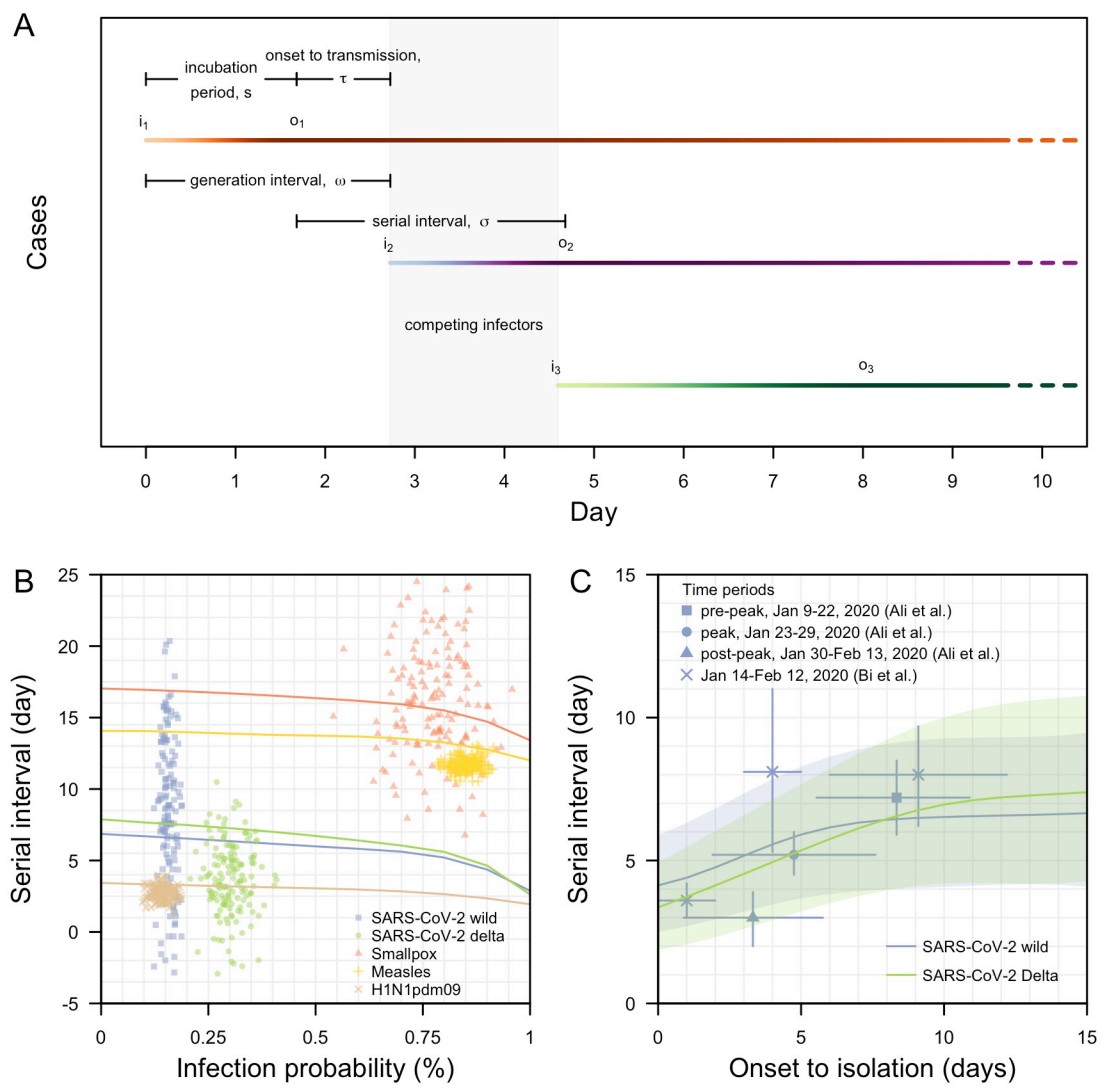

**Fig 1. Transmission dynamics of infectious diseases.** (A) Definitions of epidemiological time intervals and illustration of transmission events on calendar timescales; (B) Modelled (lines) median serial intervals for varying peak infectiousness and hence overall probability of infection for the duration of infectiousness of respective diseases. Range of observed serial interval and attack rate (a proxy for infection probability) for respective diseases in Table 1 (points) for comparison; (C) Modelled serial interval for varying delay in case onset-to-isolation in SARS-CoV-2 wild type and Delta variant with median (lines) and interquartile range (shaded regions). Observed serial intervals from published studies [12,14] as shown in points (mean) with lines (95% CI).

infectiousness profile of the infector (Fig 1A). We simulated 1,000 transmission pairs under varying pathogen biology and outbreak control measures before comparing the modelled serial intervals with those reported in real-life outbreaks to validate the modelling framework.

## Influence of pathogen biology and outbreak control measures on observed serial interval

The magnitude of any differences in generation and serial intervals depends on pathogen biology and the transmission process (Fig 1A and Table 1). At low levels of peak viral load (i.e. average probability of infection per contact pair is less than 25%), we estimated that the

**Table 1. Parameters to model the infectiousness profile of different diseases.** References in parenthesis.

| Disease | Incubation period | Start of infectiousness | End of infectiousness | Time of peak infectiousness | Secondary attack rate; used as a proxy for average probability of infection per contact pair |
|---|---|---|---|---|---|
| SARS-CoV-2 wild | Lognormal mean (log) = 1.62, sd (log) = 0.42 [25] | Gradually increase since time of infection [9,34] | 10 days post symptoms onset [27] | At symptoms onset [10,34] | 13.2%– 18.2% [27] |
| SARS-CoV-2 Delta | Weibull shape = 2.23, scale = 4.68 [35] | Gradually increase since time of infection [9,34] | 18 days post symptoms onset [36] | At symptoms onset* [9,34] | 23.0%– 37.3% [27,37] |
| Smallpox | Normal mean = 12, sd = 1 [38,39] | Upon symptoms onset [40] | 14 days post symptoms onset [41] | Three days post symptoms onset [41] | 60.0%– 90.0% [42,43] |
| Measles | Normal mean = 14, sd = 1.5 [44] | Four days before symptoms onset [44] | Four days post symptoms onset [44] | At symptoms onset [44] | 80.0%– 90.0% [44] |
| Influenza | Normal mean = 2, sd = 0.5 [45,46] | One day before symptoms onset [47,48] | Six days post symptoms onset [47,48] | At symptoms onset [47,48] | 11.0%– 18.0% [45] |

*Sensitivity analysis elaborated in section on scenarios

median serial interval decreased by less than 0.5 days as probabilities of infection per contact increased (Fig 1B and Fig A in S1 Text). These changes were small and approximately linear as the first order term in a Poisson process dominates when the force of infection is low.

At higher levels of peak viral load, we estimated that diseases with a moderate pre-symptomatic period, such as SARS-CoV-2, median serial intervals decreased by 0.7 days when the probability of infection increased from 25% to 50%; and decreased by a further 0.9 days when the probability of infection increased from 50% to 75%. However, for diseases with a short pre-symptomatic phase such as influenza, the decline in the median serial interval was 0.2 days for a probability of infection increasing from 25% to 50% and 0.3 days for an increase from 50% to 75%. Thus, the influence of peak viral loads on serial intervals is greater when viral loads are high with a longer pre-symptomatic infectiousness phase.

Comparing the simulated median serial intervals with the range of observed values in the literature, substantial variation was observed for diseases such as smallpox and SARS-CoV-2 (Fig 1B). For smallpox, this variation can be attributed to the long incubation period and duration of infectiousness. For SARS-CoV-2 wild type and the Delta variant, our analysis suggested that this variation could be attributed to changes in duration of symptoms onset-to-isolation over the course of the outbreak (Fig 1C). Thus, besides pathogen biology, serial intervals are also influenced by population-level outbreak control measures [12], which would need to be controlled for when comparing the epidemiological properties of variants.

We estimated that the difference in the median serial intervals for wild-type SARS-CoV-2 and the Delta variant would not exceed one day across a range of values for symptoms onset-to-isolation, and the interquartile range overlapped considerably (Fig 1C and Fig B in S1 Text). This suggests that there is an inherent epidemiological constraint to detect large serial interval differences for these specific variants, even under very different control scenarios. Published estimates on the serial intervals of SARS-CoV-2 wild type for respective delay in symptoms onset-to-isolation followed a broadly similar pattern to these model predictions (Fig 1C).

## Power to detect differences in generation and serial interval for transmission pairs

Any measured difference in the mean generation and serial interval based on available data depends on the statistical power of the analysis (i.e. ability to correctly detect a true difference

of a given magnitude). To estimate this power, we first need to identify the combination of biological and epidemiological characteristics that would give rise to a particular difference between a reference and alternative pathogen.

For each combination of biological and epidemiological characteristics, we simulated 1,000 transmission pairs with full knowledge on the time of events (e.g. infection, isolation). In our baseline scenario, we assumed constant outbreak dynamics (i.e. no exponential growth or decay). We then compared the difference in the means of the generation intervals between a reference and an alternative pathogen. We used Welch's t-test to compute the power to detect this difference for a given number of transmission pairs; the same steps were repeated for comparing serial intervals.

In reality, serial intervals are more commonly observed and generation intervals are inferred from these observed serial intervals. Thus, in our three-part inference process, we estimated: (i) the power to detect the theoretical difference in the generation intervals, (ii) the power to detect the observed difference in serial intervals, and (iii) the power to detect the inferred difference in generation intervals.

## Different incubation period between variants

As a case study, we modelled the reference and alternative pathogen to have a similar peak viral load and duration of shedding post-peak viral load as the Delta- and Alpha-like variants, then extracted the combinations of parameters that gave rise to a one-day reduction in the generation interval of the Delta-like pathogen. Under a scenario of either no isolation or mean symptom onset-to-isolation of 8 days, we estimated the incubation period would need to be 1.6 days shorter for the Delta-like variant to generate a one-day shorter generation interval. When the mean symptom onset-to-isolation was 4 days, the corresponding incubation period was 1.4 days shorter to generate a one-day difference in generation interval (Fig 2A).

For a sample size of 100 transmission pairs, we used these extracted characteristics to calculate the corresponding theoretical power to detect a one-day difference in the generation interval. We estimated the power was 32% with no isolation in place, 48% with onset-to-isolation of 8 days and 66% for 4 day delay from onset-to-isolation. As the onset-to-isolation time decreases, the power to detect differences in generation interval increases due to the reduced variance in the generation interval distributions of the reference and alternative pathogens (Fig C in S1 Text). Because more transmission events in the tail of the distribution are being prevented with a rapid onset to insolation, more of the 100 sampled transmission pairs come from samples near the mean.

As a sensitivity analysis, we also modelled the infectiousness profile using a function derived from the analysis of observed wild type SARS-CoV-2 transmission pairs by Ferretti et al [9]. The differences in incubation period for a one day shorter generation interval in the model by Ferretti et al was comparable to our modelled results (Fig D in S1 Text).

When we performed the same analysis comparing serial intervals between variants, we found that differences in the incubation period for a one-day shorter serial interval in the Delta-like variant were similar to the findings for the generation intervals. However, the power to detect a one-day difference in the serial intervals was lower. For a sample size of 100 transmission pairs, it was 29% with no isolation, 40% with onset-to-isolation of 8 days and 54% for onset-to-isolation of 4 days (Fig 2B). Unlike generation intervals, serial intervals are a combination of biological quantities in two different individuals: the incubation period in the infectee and the onset-to-transmission of the infector (Fig 1A); these quantities were assumed to be biologically independent in our analysis. On the contrary, the generation interval depends only on the infector's delay from infection-to-transmission. This typically results in a lower

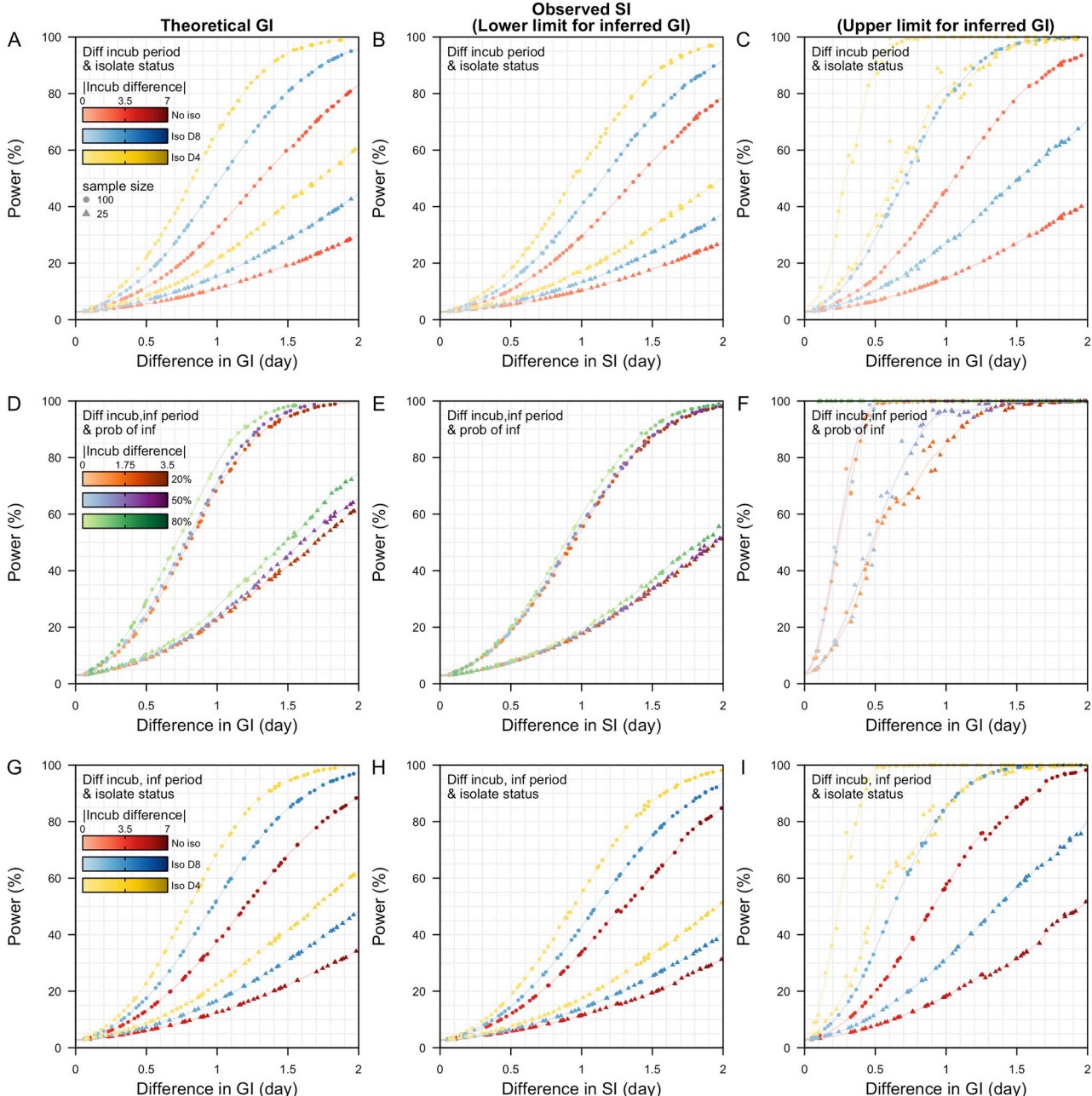

**Fig 2. Power to detect differences in the generation intervals (GI), serial intervals (SI) between reference and alternative pathogen.** (A-C) Different incubation period between reference and alternative pathogen under the same symptoms onset-to-isolation status of either no isolation, mean symptoms onset-to-isolation of 8 days, or 4 days; (D-F) Different incubation period and longer duration of infectiousness post-peak viral load in reference pathogen under the same mean symptoms onset-to-isolation of 4 days. Peak viral load in reference pathogen is varied resulting in a probability of infection, p, of either 20%, 50% or 80% when the mean incubation of the reference pathogen was 4 days; (G-I) Different incubation period and longer duration of infectiousness post-peak viral load in reference pathogen under respective onset-to-isolation. (A,D,G) Theoretical power to detect differences in GI, (B,E,H) power to detect differences in observed SI—lower limit estimates of the theoretical power, (C,F,I) upper limit estimates of the theoretical power.

variance for the generation interval distribution as compared to serial interval distribution [15] and, hence, more statistical power.

As generation intervals are rarely observed, serial intervals are often used as a proxy for the delay between generations of infection. In general, the mean of the inferred generation interval is similar to the mean of the observed serial interval when the infector and infectee have the same incubation period distribution [15,16]. The variance of the inferred generation interval is dependent on the variance of the serial intervals and on the covariance of the onset-to-transmission and incubation period of the infector. To understand how the changes in the variance affect the power to detect differences in the inferred generation intervals, we therefore explored two extreme scenarios based on [15].

At one extreme, we assumed that the infectiousness of an infector is dependent on the time of symptoms onset only. Under this assumption, the incubation period of the infector and the time from onset-to-transmission are independent and their sum equates to the generation interval. For the same incubation period distribution in the infector and infectee, this assumption implies that the generation interval is the same as the serial interval (sum of time from onset-to-transmission in infector and incubation period of infectee). Thus, the corresponding power to detect the difference in the inferred generation and observed serial intervals is equivalent (Fig 2B).

At the other extreme, we assumed that the infectiousness is dependent on the time since infection of the infector. As such, the time of transmission is not correlated with the time of the time of symptom onset in the infector, and the variance of the derived generation interval is lower than the observed serial interval. The two assumptions serve as the upper and lower limits to the variance of the inferred generation interval (i.e. lower and upper limits of the power). Under the second assumption, the power to detect a one-day difference in the generation interval for a sample size of 100 was 46% with no isolation, 75% with onset-to-isolation of 8 days and 100% for onset-to-isolation of 4 days (Fig 2C). These power values are higher than under the assumption of the lower limit for GI (Fig 2B), but in practice this could also increase the chance of a false positive (i.e. concluding a difference in GI when there is not none).

## Different incubation period, peak infectiousness and duration of infectiousness

Variants of SARS-CoV-2 can differ by more than one biological characteristic and different combinations of these characteristics can produce similar differences in the generation and serial intervals. To explore these interactions, we modelled the reference Delta-like variant to have a longer duration of viral shedding (8 days longer) with same or higher peak infectiousness as compared to the alternative wild type-like pathogen for a range of incubation periods. We did not vary the mean symptoms onset-to-isolation delay (4 days) in order to study the reduction in the generation and serial interval arising from variation in pathogen characteristics only. For the same peak infectiousness (i.e. per-contact probability of transmission for the Delta-like variant equal to 20% with a mean incubation period of 4 days), the incubation period was 1.9 days shorter for the Delta-like variant to give a one-day shorter generation or serial interval. On the contrary, when the peak infectiousness of the Delta-like variant was higher, resulting in a 50% or 80% probability of infection, the corresponding incubation period was 1.3 days or 0.2 days shorter. The theoretical power to detect a one-day difference generation interval was between 70–85% in all three scenarios (Fig 2D). Similar differences in incubation period resulted in a one-day difference in the serial intervals and the corresponding power was about 50–65% (Fig 2E). This serves as the lower limit estimate of the power to detect the same one-day difference in the generation intervals inferred from the observed serial

intervals while the upper limit estimates was 100% (Fig 2F). The differences in the incubation period were more pronounced under scenarios of no case isolation (Table A in S1 Text). Even if we account for additional variability in the time of peak infectiousness, allowing it to occur 1–5 days post symptoms onset, the incubation period of the Delta-like variant was1.3–1.5 days shorter for a 20–50% probability of infection (Table B in S1 Text).

### Different incubation period and duration of infectiousness

Besides intrinsic differences in biological properties among variants of SARS-CoV-2, vaccination could also shorten the duration of viral shedding and, hence, infectiousness, in vaccinated cases as compared to unvaccinated cases. By modelling the mean duration of infectiousness in an unvaccinated case to be 8 days longer, for a one-day shorter generation interval in the unvaccinated cases, we estimated the incubation period was 5 days shorter in the unvaccinated cases under no case isolation; 2.9 days shorter when the mean symptom onset-to-isolation was 8 days; 1.9 days shorter when the mean onset-to-isolation was 4 days. The corresponding theoretical powers to detect a one-day difference in the generation interval were 37%, 52% and 69% respectively (Fig 2G). As compared to previous scenarios (Fig 2A and 2B), a shorter incubation period in the reference pathogen counteracts the longer shedding profile and narrows the difference in the mean generation interval of both pathogens. The reduction in the incubation period (e.g. 5 days shorter) does not necessarily correspond to the increase in duration of shedding (e.g. 8 days longer).

### Different contact pattern among household and non-household pairs

Based on measured contact patterns, we also found that frequent contacts between household members can result in higher probabilities of infection and earlier infections as compared to non-household contacts for the same pathogen characteristics. For a one-day shorter generation interval among household contacts, the difference in the probability of infection between household and non-household contacts in our baseline scenario was 57% (59% vs 2%) under no isolation; when the mean onset-to-isolation was 8 days, this difference was 65% (68% vs 3%); when the symptoms onset-to-isolation was 4 days, the difference was 74% (78% vs 4%). For a sample size of 100 transmission pairs, the theoretical power to detect a one-day difference in the generation intervals were 33%, 51% and 68% respectively (Fig 3A). Based on previous literature, the probabilities of infection in household pairs are typically estimated to be less than 50% (18, 20). For a 20–40% probability of infection in household contacts, the probability of infection in non-household contacts was 0.6–1% in our analysis. We estimated that the differences in generation and serial intervals among such contacts were 0.2–0.4 days and the corresponding power to detect these differences in the generation and serial intervals were less than 1% (Fig 3A–3C).

### Different contact frequency between household pairs

When the frequency of contact is low (e.g. weekly household-type contacts), the timing of the contacts matters more as it determines which portions of the infectiousness period (e.g. start or end) would have the highest concentration of the limited infection opportunities. For the same pathogen, under no case isolation, the frequency of contact can therefore have a considerable impact on transmission risk. In a scenario where the probability of infection was 20% among household members who had daily measured contact with an infectious individual, the corresponding probability of infection among individuals who had only weekly household contacts dropped to 2%. We estimated that the mean generation and serial intervals were 1.0 and 1.1 days shorter for daily household contacts when there is no case isolation. For a sample

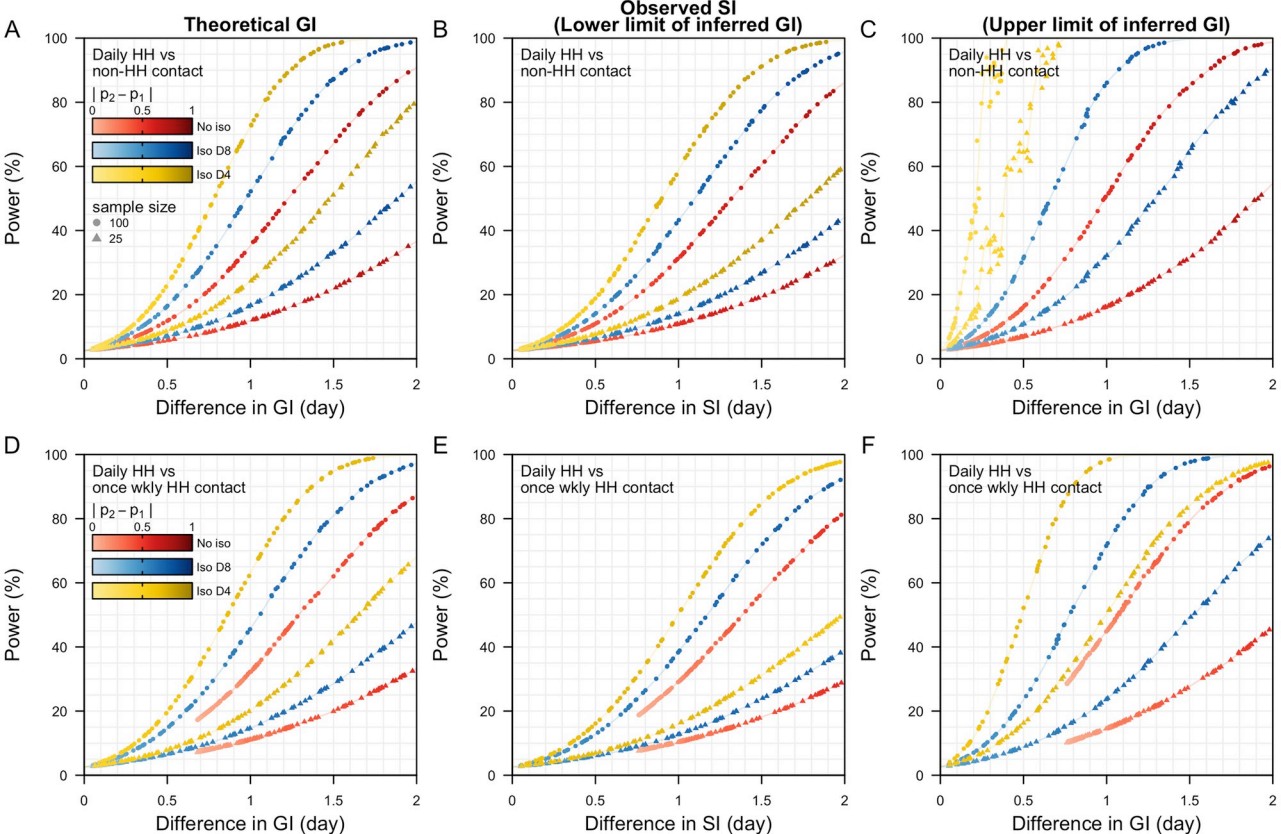

**Fig 3. Power to detect differences in the theoretical generation intervals (GI), observed serial intervals (SI) and derived GI between reference and alternative pathogen.** (A-C) Different contact patterns of either non-household or household contact but same incubation period under respective peak infectiousness and isolation status. Due to the differences in contact frequency, probabilities of infections ($p_1$ for reference pathogen and $p_2$ alternative pathogen) are different for both types of contact for the same peak infectiousness; (D-F) Different contact patterns of either daily or weekly household contact but same incubation period under respective peak infectiousness and isolation status. (A,D) Theoretical power to detect differences in GI, (B,E) power to detect differences in observed SI—lower limit estimates of the theoretical power, (C,F) upper limit estimates of the theoretical power.

size of 100, the theoretical power to detect the differences in the generation and serial intervals was 33% (Fig 3D and 3E). When the mean duration of symptoms onset-to-isolation was 4 days, the corresponding probability of infection was 64% among household members with daily contact, and 7% for those with weekly contact. The mean generation interval was 0.3 shorter for daily household contacts while the serial interval was 0.2 days longer. The power to detect these differences in both intervals was less than 10% (Fig 3D–3F).

## Different epidemic growth dynamics of variants

When a new variant is introduced in a population, the growth rate of this variant and the existing pathogen may differ. Under exponential growth dynamics, the newly observed cases are more likely to be recently infected with shorter incubation periods. The overall incubation period in the population without adjusting for these dynamics will be shorter [11] and potentially bias the measured generation and serial intervals when left unadjusted. Thus, we need to understand the magnitude and direction of this bias.

We modelled a scenario where the reference pathogen has a one-day shorter incubation period but a longer duration of viral shedding (i.e. Delta-like) as compared to the alternative

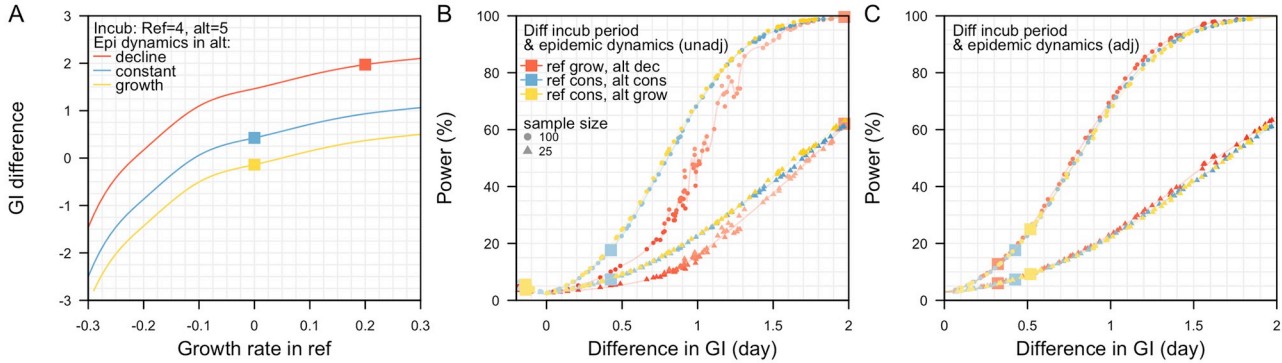

**Fig 4. Generation intervals, GI, under varying outbreak dynamics.** (A) Differences in the generation interval between the reference and alternative pathogen without adjusting for exponential growth or decline outbreak dynamics. Exponential growth of 0.2/day (red), constant outbreak (blue) and exponential decline of 0.2/day (yellow) in alternative pathogen, (B) corresponding power to detect the biased differences in the generation intervals, (C) power to detect differences in the generation intervals after correctly adjusting for exponential outbreak dynamics.

pathogen (i.e. wild-type-like). We also varied the epidemic dynamics in each pathogen. When there was constant growth in both pathogens, the observed mean generation interval was 0.4 days shorter in the reference pathogen. In the absence of bias from the epidemic phase, the true interval would therefore be 0.4 days. When there was exponential growth of 0.2 per day in the reference pathogen and exponential decline of 0.2 per day in the alternative pathogen, the resulting mean generation interval–which is influence by the combined epidemic process and incubation period distribution–was 2.0 days shorter for the reference pathogen (Fig 4A). Under a scenario of constant growth in the reference pathogen but exponential growth of 0.2 per day in the alternative pathogen, the observed mean generation interval was 0.1 days longer in the reference pathogen. Hence in any analysis, we must simultaneously consider differences in true generation time and bias from the epidemic phase.

The power to detect a difference in the generation intervals is dependent on the extent of overlap in the generation interval distribution of the reference and alternative pathogen. The extent of this overlap is in turn affected by the differences in the biological characteristics of each pathogen. However, this overlap also depends on the prevailing outbreak dynamics. Without adjusting for exponential growth and decline dynamics in the reference and alternative pathogen respectively, the extent of overlap in the generation interval distributions of the reference and the alternative pathogen is lesser as compared to the scenario where both pathogens are at constant incidence (Fig E in S1 Text). This accentuates the differences in the mean generation interval, thereby increasing the power to detect this difference and conclude that there exist non-zero difference between the generation intervals of two pathogens (i.e. lower Type II error) (Fig 4B, red square). Furthermore, unadjusted outbreak dynamics can also increase the chance of concluding a difference in the generation intervals when there is none after adjustment (i.e. higher Type I error). However, when we correctly adjust for the exponential outbreak dynamics, we recovered a similar mean difference in the generation intervals of the reference and alternative pathogen across different combinations of epidemic dynamics, and a similar power to detect this difference (Fig 4C).

Overall, the sample size of a study should be designed based on the desired power to detect a difference after adjusting for the observed epidemic dynamics. Without adjusting for epidemic dynamics, there is a possibility of accentuating the differences in the generation intervals for a reference pathogen undergoing exponential growth and an alternative pathogen experiencing exponential decline. Consequently, even with a small sample size there is a large

power to detect this biased difference (Fig 4B). However, after adjusting for the epidemic dynamics, the power to detect the inherent difference in the generation intervals would be reduced (Fig 4C). Thus, it is important to account for epidemic dynamics when planning for the appropriate number of samples for collection under the prevailing or likely outbreak dynamics.

## Generation and serial intervals in households with multiple competing infectors

Within household outbreaks, infectors compete for the remaining susceptible individuals, which can influence the dynamics of the observed transmission events in a cluster. For each cluster, we simulated two transmission pairs involving three individuals; the first pair was an index and a secondary case, the second pair was either a secondary and a tertiary case, or the index and another secondary case. We simulated 1,000 clusters and estimated the distribution of the generation and serial intervals over different onset-to-isolation delays.

We estimated that transmission events involving multiple competing infectors resulted in a lower median generation interval as compared to pairwise transmission involving a single infector. The magnitude of this difference increases when the median delay from onset-to-isolation increases (Fig 5). For an assumed mean incubation period of 4 days and onset-to-isolation of 4 days (variance 5 days), the corresponding time from symptoms onset to transmission was 0.9 days (95%CI -3.4–5.8) in pairwise transmission but 0.4 days (95%CI -3.9–5.2) for cluster transmission. Taking the difference between the pairwise and cluster transmission, the mean difference in the generation interval distribution was 0.4 days (95%CI 0.2–0.7) (Fig 5B). When the delay from onset-to-isolation was 8 days, the difference in the time from symptoms onset-to-transmission between pairwise and cluster transmission widened and the mean difference in the generation interval distribution increased to 0.7 days (95% 0.4–1.0). When exposed to multiple infectors, the probability of a susceptible individual being infected in a timestep given no previous infection would increase and, hence, reduce the expected time until infection.

The overall incubation period distribution in the modelled transmission events in households with multiple infectors was similar to that in pairwise transmission. In both pairwise and cluster transmission, we modelled the mean incubation period of all primary cases as 4 days (variance 5 days). The mean incubation period of secondary cases with onward transmission

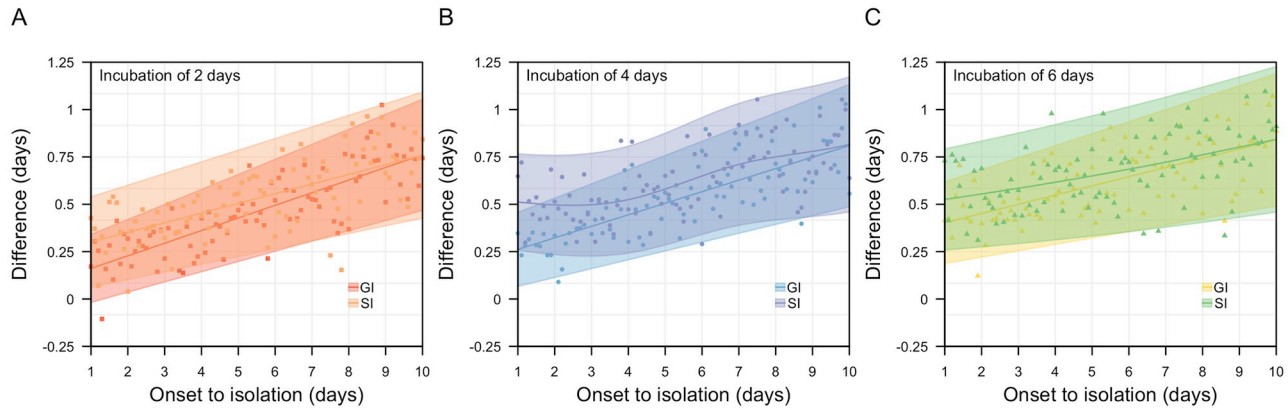

**Fig 5. Differences in mean generation (GI) and serial (SI) intervals for transmission between pairs (i.e. no competing infector) and triples (i.e. competing infectors) with mean incubation period of (A) 2 days, (B) 4 days and (C) 6 days.**

was 4.2 days (variance 5.3 days) and the difference in the incubation period between different generations of infectors was 0.2 days (95%CI -0.5–0.03) when infectors were isolated on average 4 days after symptoms onset. Similar outcomes were observed when the delay from onset-to-isolation of infectors was increased to 8 days. As such, while secondary cases with short incubation periods experience earlier onset and peak viral load as compared to the primary cases, they were not observed to transmit more infections to the third susceptible individual to shorten the overall mean incubation period. As the generation and serial intervals are a combination of the time from symptom onset-to-transmission and the incubation period, the shortening of these intervals in a cluster transmission is largely driven by the reduction in the onset-to-transmission rather than the incubation period.

## Discussion

Using a model incorporating high resolution human interactions, we found that interacting biological and epidemiological processes can have a major impact on ability to detect changes in observed pathogen generation and serial intervals. Using novel SARS-CoV-2 variants as a case study, we showed that statistical power to estimate differences in the generation or serial intervals between variants can be highly sensitive to factors such as the incubation period and delay from onset-to-isolation. With a large sample size of 100 transmission pairs, the power of studies to detect a one-day change in the generation interval can be 30–70% depending on the prevailing delay from onset-to-isolation. This power could decline to less than 20% when the sample size is reduced to 25 transmission pairs.

Assuming either a linear or exponential relationship between the generation interval and growth rates [17], if the generation interval decreases by one day (e.g. from 5 days to 4 days), this could result in a 25% increase in growth rate with a reproduction number of 2. In other words, if we compare the initial growth dynamics of the old and new variant, the outbreak trajectory in the latter will double that of the former in about 3 weeks based on the changes in generation intervals only. For SARS-CoV-2 with an initial generation time of about 5–6 days [18,19] and doubling time of 2–4 days [20], countries have reported to take about 1–3 weeks to expand isolation facilities or testing capacity by at least 2 times at the start of the outbreak [21–23]. Thus, when faced with a novel faster variant, early studies to detect changes in generation intervals and, hence, growth rates, may be underpowered. Furthermore, the timescales for expanding healthcare capacity is potentially slower than the outbreak growth rate. Overcoming these challenges would require implementation of strict population-level outbreak control measures (e.g. physical distancing, mask-wearing) to slow the outbreak at the initial phase, to buy time to expand the healthcare capacity and gather information on the new variant.

Studies with small sample size of 30–50 transmission pairs are likely to be underpowered to detect small differences of 1–2 days in the generation or serial intervals [7,24] but our simulation framework allowed us to explore these differences in the absence of biases created during the data collection process. We showed that when the probability of infection is 20–50% and the delay from symptoms onset-to-isolation is 4 days, the corresponding incubation period of the Delta variant would need to be shortened by 1.3–1.9 days to observe a one day shorter serial interval. When there is no case isolation, a larger difference in the incubation period was required to achieve the same effect. Direct comparison of the incubation periods from different studies suggest that the incubation period of the Delta variant is 0–1.4 days shorter than the wild type SARS-CoV-2 [15,24–26] and the secondary attack rate of Delta (proxy for probability of infection) ranges from 23.0–37.3% [27]. Taking into consideration the findings from other studies and our modelling outputs, this suggests that a shorter serial interval of at least one day is not likely to occur between the wild-type SARS-CoV-2 and the Delta variant.

Outcomes from our modelling framework are comparable with other epidemiological studies. The mean generation and serial interval of the Delta household transmission pairs were estimated to be 0.7 days and 1.7 days shorter than that of the Alpha variant [6]. In that study, the mean incubation period of the Delta cases was estimated to be 1.4 days shorter. Based on our modelling framework, the incubation of the Delta variant needs to be about 1–2 days smaller for a one-day reduction in the generation or serial interval, assuming the duration of infectiousness and the peak infectiousness of both variants are similar. Empirical findings from different countries and regions also reported an incubation period of about 4 days for the Delta variant at different time points of the outbreak; a day shorter than the estimated incubation period of the Alpha variant [28,29]. For the same pathogen but different contact frequencies, we estimated small differences in the serial intervals of less than half a day when the probabilities of transmission in non-household members are small. This corroborates with one study estimating an empirical difference of less than 0.5 days between household and non-household members during the peak of the COVID-19 pandemic involving the wild-type SARS-CoV-2 in China in Jan 2020 [12] with a household attack rate of about 20% (a proxy for probability of infection) [30].

While generation and serial interval distributions are shortened due to a decrease in the mean incubation period during an exponential growth phase of an outbreak [11], the occurrence of multiple infectors in a household transmission cluster can also reduce these intervals. This reduction occurs when the time from symptoms onset to infection is shortened. In a modelled cluster with competing infectors, infectors with shorter incubation periods were not observed to preferentially transmit infection to the susceptible individual. Differentiating the reasons for faster outbreak growth is important. If the growth of an outbreak is driven by a true biological reduction in the incubation period, the outbreak control policy would need to focus on rapid and wide contact tracing beyond the household. Exponential growth dynamics may bias our interpretation of the change in a pathogen's incubation period and, hence, changes in the generation and serial intervals but appropriate adjustments would rectify the bias. On the contrary, if the growth of an outbreak arises from increase in earlier household transmission, especially during periods of lockdown, control policy would then need to shift towards effective household isolation.

There are some limitations to our study. Firstly, we did not explore the effect of viral dose exposure on the probabilities of transmission over a contact [31]. The duration of a contact can be long and continuous or occur as a series of short contacts with breaks in between. For a continuous contact episode (i.e. a series of consecutive 5-minute contact records), we assumed that the force of infection is summative across the timesteps and constrained the probability of infection among household contacts over the entire period of infectiousness to match the observed secondary attack rates in households. The lack of in vivo studies on transmission probabilities over contact duration poses a challenge to evaluate dose-response relationships but could be explored in future simulation studies.

Secondly, we did not account for variations in the start and end times of the infectiousness profile, and instead fixed these parameters based on average observed durations of viral growth or decline. Furthermore, the scale factor for the peak infectiousness were not modelled based on a distribution. Accounting for these variations are not likely affect the mean difference in the generation and serial intervals or the parameters (e.g. mean incubation period) that result in this difference but will lead to reduced power to detect these differences. We estimated the power to detect a difference in the generation intervals with a Welch t-test using the estimated serial intervals for the reference and alternative pathogen. We considered two bounding assumptions about the population-level relationship between the variance of the generation intervals and serial intervals to obtain a plausible range of power values. However, in reality,

the inference method for obtaining generation interval distributions could introduce additional uncertainty. If we were to instead try and infer this relationship from individual transmission pairs, it would be important to account for the resulting parameter uncertainty to avoid underestimating the variance of the distributions and, hence, the power.

Thirdly, due to data identifiability issues, the relationship between a pair of contacts used in this dataset was not available and we have made a conservative assumption that contact signals within 10m translate to an effective contact. A greater (smaller) radius of detection, would generally lead to more (less) contact episodes between a pair of individuals. We would then expect the scale factor for the peak infectiousness to decrease (increase) in order to achieve the same overall probability of infection for a given range of observed attack rates for a disease. We expect the trends in the overall findings to remain similar but the use of real-world temporal networks with a well-defined edge list between individuals would refine the analysis.

Standardising the contact patterns and effects of non-pharmaceutical interventions to compare changes in pathogen biology and, hence, changes in generation and serial interval in outbreak data is challenging. By simulating known changes in the disease-related factors (e.g. incubation period and duration of infectiousness) and other external factors, we studied how sensitive these intervals were to respective factors. Based on the combination of multiple factors and measured quantities, this helps to clarify contradictory outbreak observations, evaluate the power of detecting such observations and inform future data collection efforts to ensure that studies are well powered.

## Materials and methods

### Ethics Statement

The study was approved by the London School of Hygiene & Tropical Medicine Observational Research Ethics Committee (ref. 25727). All data and analysis were collected and performed in line with the Infectious Diseases Act in Singapore which permits the collection and publication of surveillance data.

**Contact data.** Previously published social contact data recorded interactions among 469 participants over three consecutive days (Thursdays 12 Oct–Saturday 14 Oct, 2017) from 0700–2300 hours each day, as part of the BBC Pandemic study conducted in Haslemere, United Kingdom [13] (Fig F in S1 Text). In the previous published study, participants consent to the collection of their contact data when they downloaded the BBC Pandemic mobile phone application for the purpose of that study. Using secondary data for our analysis, we defined a contact to exist between two individuals when there was a recorded signal in either of their BBC Pandemic mobile phone applications with a GPS distance of at most 10 metres apart in a 5-minute interval. Familial and friendship status were not available in the published individual-level data to avoid re-identification. Thus, we assumed that likely household contacts were represented by pairs of individuals with at least one recorded contact in five out of the six time periods from 0700–0800 hours or 2000–2300 hours over the three days. These time periods are beyond the typical working hours on weekdays before the COVID-19 pandemic [32] and consistency of contact over three consecutive days was assumed to rule out non-household contacts (e.g. commuting) occurring by chance. Based on these assumptions, we identified 54 households with an average size of 2.3, similar to previous survey estimates on household sizes in Haslemere [33], and there were 82 household and 451 non-household contacts.

As the infection process for SARS-CoV-2 typically occurred on timescales lasting more than three days [2,6], we extended the contact between a pair of individuals by randomly sampling their daily contacts over weekdays based on the recorded contact patterns on Thursday and Friday and fixed all weekend contacts based on Saturday. This process was applied to both

household and non-household contacts. To study the transmission over once-weekly interactions (e.g. weekly events or meetings), we randomly sampled a day of the week and set all contacts in other days to null.

**Infectiousness profile.** For each individual, we simulated the start and end time of the infectiousness period, with time of peak infectiousness for respective diseases relative to the incubation period (Table 1). The relative infectiousness ranged from 0 to 1—normalised relative to the peak infectiousness. We then fitted a cubic Hermite spline through the start, peak and end points of the infectiousness period. We constrained the slope of the spline to be zero at each of the three points (i.e. first derivative is zero) to simulate the infectiousness profile over the course of the infection. Furthermore, we scaled the splines of respective disease such that the probability of infection matches observed data (Table 1).

We used the SARS-CoV-2 Delta variant infectiousness profile in the main analysis to compare differences in the generation and serial interval distributions under changing pathogen biology, contact patterns and outbreak response. For sensitivity analysis, we used the skew-logistic model by Ferretti et al [9] to compare with the findings from our wild type SARS-CoV-2 spline model. This alternative model concurrently estimates different components of an infectiousness curve (e.g. growth, decline and peak) from observed wild type SARS-CoV-2 transmission pairs. It assumes a long-tail at the start of the curve for a pathogen with a long incubation period resulting in a longer pre-symptomatic infectious period. However, the model was not updated for subsequent variants. Conversely, the spline model allows for easy parameterisation of each component of the infectiousness curve based on a variant's characteristics derived from separate studies.

**Simulating transmission.** We simulated the infection of a susceptible individual through a Poisson contact process. In each 5-minute interval contact episode, $t$, the conditional probability of infection given no prior infection, $p_{inf}(t)$, is defined as:

$$p_{inf}(t) = e^{-\Lambda(t-1)}(1 - e^{-\lambda(t)})$$
$$\approx \lambda(t) \tag{1}$$

$$\lambda(t) = \beta v(t)c(t)h(t) \tag{2}$$

$$\Lambda(t-1) = \sum_0^{t-1} \lambda(t) \tag{3}$$

where $\lambda(t)$ is the force of infection and is a function of the relative infectiousness, $v(t) \in [0,1]$, scaled by a factor $\beta$ to constrain the overall probability of infection to be similar to the observed attack rate; the presence or absence of contact between two individuals, $c(t) \in \{0,1\}$, and the current isolation status of the infector, $h(t) \in \{0,1\}$. $\Lambda(t)$ represent the cumulative force of infection up to time $t$. The first coefficient on the RHS of Eq 1 is the probability of surviving infection up to time step $t-1$ and the second coefficient is the probability of being infected at time step $t$. The stochastic model then samples the time of infection in each pair of individuals based on $p_{inf}$. For small values of $\lambda(t)$, Eq 1 approximates to $\lambda(t)$.

Each contact pair has a unique sequence of recorded signals (Fig F in S1 Text) and a corresponding cumulative probability of infection in all 5-minute interval contact episodes over the entire infectiousness period of the infector (i.e. probability of infection per contact pair). We defined the probability of infection to be the average probability of infection per contact pair. Once the simulated transmission occurred between a pair of individuals, there was not further propagation of the infection. We simulated 1,000 transmission pairs for each combination of pathogen and epidemiological characteristics.

Under a scenario of 'competing infectors', we simulated two susceptible individuals being exposed to an index case. Pairwise transmission was modelled and after the first transmission event had occurred, the remaining susceptible individual would subsequently be exposed to an additional infector, thereby acquiring infection from either infectors (Fig 1A). This is similar to disease transmission in households and we assumed that all susceptible household members were only exposed to infected cases within the household. Intuitively, infectors with a shorter incubation period are more likely to have earlier infectious contact with existing susceptible household members and thus potentially resulting in shorter generation intervals, over the generations. However, infectors with longer incubation periods tend to have longer pre-symptomatic infectious periods [9,49] and, for the same duration of shedding post symptoms onset, these infectors potentially exert a higher force of infection on the susceptible individuals over the entire duration of infectiousness which could influence the generation intervals over the generations. Thus, we investigated the differences in the generation and serial interval distributions, for transmission in pairs and triples.

**Scenarios.** For each disease in Table 1, we investigated how variations in the scale factor for peak infectiousness, $\beta$, would change the probability of infection and serial interval using the spline model. Furthermore, for the wild-type SARS-CoV-2 and the Delta variant, we studied how variations in the delay from symptom onset-to-isolation would vary the serial interval. All transmission events were simulated using household member contact patterns (unless otherwise stated) to achieve a similar probability of infection as the observed secondary attack rate in households. Model outputs were compared against published data on serial intervals and attack rates to ensure they were within the observed range.

**Pairwise transmission.** In the main analysis on pairwise transmission, the incubation period and infectiousness profile of the reference pathogen were based on SARS-CoV-2 Delta variant (Table 1). We compared how different scenarios of changing pathogen biology would influence the changes in the generation and serial intervals, and the corresponding power to detect these differences under varying human contact patterns and outbreak responses. Namely, we studied the effects of the following responses: no isolation, average delay of 4 days from symptom onset-to-isolation, average delay of 8 days from symptom onset-to-isolation (Table 2). As a sensitivity analysis, we assumed that the time of peak infectiousness occurred 1–5 days after symptoms onset [49,50] instead of at the time of symptom onset (Table 1).

**Epidemic dynamics.** We first studied the difference in the generation interval distribution between a reference and alternative pathogen without adjusting for the bias introduced by varying epidemic dynamics. The reference pathogen had a one-day shorter incubation period and longer duration of viral shedding (i.e. Delta-like) as compared to the alternative (i.e. wild-type-like). During exponential growth, the incubation periods of the recently infected infectors (also known as the backward incubation period in [5,11]) tend to be shorter than the true incubation period (also known as the forward incubation period in [5,11]). Without adjusting for the outbreak dynamics, the overall observed incubation period will be shorten and, hence, shortening the generation intervals. The relationship between the observed (backward) and the true (forward) incubation period can be expressed as [5]:

$$b(\tau) = \frac{exp(-r\tau)f(\tau)}{\int_0^\infty exp(-rx)f(x)dx} \tag{4}$$

where $b(\tau)$ is the backward incubation period and $f(\tau)$ is the forward incubation period $\tau$ time since infection, and $r$ is the exponential growth rate if $r > 0$ and exponential decline if $r < 0$. We parameterised our model by $f(\tau)$ to generate the $b(\tau)$ of the infectors and simulate the

**Table 2. Simulated scenarios and how they relate to observations in the SARS-CoV-2 pandemic.**

| Scenario | Observations |
|---|---|
| Different incubation period between reference and alternative pathogen but same peak infectiousness and duration of shedding post-peak infectiousness for respective symptom onset-to-isolation. | SARS-CoV-2 Delta and Alpha variants were reported to have similar peak viral load and duration of shedding after the peak [2] with the former having a shorter incubation period [28]. |
| Different incubation period between reference and alternative pathogen for respective symptom onset-to-isolation. Peak viral load in reference pathogen was varied ($\beta$ of 0.0005, 0.002, 0.006) resulting in either a 20%, 50% or 80% probability of infection when the mean incubation of the reference pathogen is 4 days. Peak viral load in alternative pathogen was fixed ($\beta$ of 0.0005). Duration of shedding post-peak infectiousness for the reference pathogen was 8 days longer. | SARS-CoV-2 Delta variant was reported to have a longer duration of viral shedding than the wild type but contrasting findings were reported for differences in peak viral load and incubation period [1,3,26]. Furthermore, some studies conducted during the exponential growth phases might not be explicitly adjusted for recent infections (i.e. increased observations of cases with short incubation periods). As such, contrasting findings of serial intervals shortening by 1–2 days [24,26] or no change in these intervals after accounting for earlier case isolation [7,8] were reported. |
| Different incubation period and shorter duration of infectiousness post-peak viral load in the alternative pathogen under respective symptom onset-to-isolation. Duration of shedding post-peak infectiousness for the reference pathogen was 8 days longer. | Similar peak viral load was reported in both vaccinated and unvaccinated SARS-CoV-2 Delta cases, but the former has a shorter duration of shedding post peak viral load [1,2,36]. Small sample size and infrequent data points for viral growth trajectories as compared to viral decline affected the power to detect differences in viral growth rates [1,2]. This difference, if any, would suggest a different duration of shedding prior to peak viral load thereby affecting the ability to detect cases early and the extent of pre-symptomatic transmissions. |
| Different contact patterns of either non-household or household contact with the same peak infectiousness and same incubation period under respective symptom onset-to-isolation over a range of peak infectiousness. Due to the differences in contact frequency, probabilities of infections ($p_1$ and $p_2$) were different for both types of contact. | Large scale movement restrictions such as lockdowns and work-from-home arrangements would potentially increase the proportion of contacts occurring with household members among all contacts and a corresponding decrease for non-household contacts [51,52]. Thus, for the same pathogen characteristics, the frequent contact in households would alter the probability of infection in each timestep as compared to non-household contacts, thereby altering the generation and serial intervals. |
| Different contact patterns of either daily or weekly household contact but with the same peak infectiousness and same incubation period under respective symptom onset-to-isolation over a range of peak infectiousness. | |

different outbreak dynamics. We varied the exponential rate *r* in the reference pathogen from -0.3 to 0.3 in increments of 0.1. We also varied the *r* in the alternative pathogen to be 0.2, 0 and -0.2 which corresponds to an outbreak with doubling time of 3.5 days, sustained constant outbreak, and half-life of 3.5 days.

In a real-world outbreak, with a given backward incubation period and a known exponential rate, we can adjusted for the bias brought about by the exponential outbreak dynamics and derive the forward incubation period as follow [5]:

$$f(\tau) = \frac{exp(r\tau)b(\tau)}{\int_0^\infty exp(rx)b(x)dx} \quad (5)$$

To illustrate the changes in power to detect the mean difference in the generation interval, we simulated a scenario of varying incubation period for a reference and alternative pathogen under (i) exponential growth in a reference pathogen but exponential decline in the alternative pathogen, (ii) constant growth in both reference and alternative pathogen, (iii) constant growth in the reference pathogen and exponential growth in the alternative pathogen.

**Cluster transmission.** For transmission occurring under competing infectors, we simulated 1000 transmission clusters and investigated the changes in the generation and serial intervals for transmission in a cluster of triples as compared to the previous pairwise transmission. We varied the incubation period with an average of 2, 4, or 6 days, under a $\beta$ value of 0.0005 (scale factor to achieve similar peak infectiousness as SARS-CoV-2 Delta variant) for different delays from symptom onset-to-isolation.

**Statistical test.** We used a two sample Welch's t-test to estimate the power to detect a difference in means of the interval distributions of the reference and alternative pathogen. The test was two-sided with a significance level of 5%. An equal number of intervals were sampled from each distribution (25 or 100 samples out of 1,000 simulated pairs) and distributions are of unequal variances.

In reality, generation intervals are rarely observed and would need to be inferred using the observed serial intervals [15]. We studied two inference approaches: that the infectiousness of an infector is (i) dependent on the time of symptoms onset of the infector or (ii) dependent on the time of infection of the infector [15]. In each approaches, we would infer the mean and variance of the generation interval (i.e. inferred generation interval) based on the assumptions elaborated in [15] and S1 Text. Using this mean and variance, we will perform the Welch's t-test to obtain the estimated power to detect a difference in the means of the inferred generation interval distribution of the reference and the alternative pathogen.

In both approaches, we assumed that the incubation period distribution of the infector and the infectee are independent and identically distributed. As such, the inferred generation interval and observed serial interval have the same mean. The relationship between the variance of the inferred generation interval can be expressed broadly as follows [15] with further elaboration in Text A in S1 Text:

$$Var(G) = Var(S) + 2Cov\left(P_{ij}, I_i\right) \tag{6}$$

where $G$ is the generation interval, $S$ is the serial interval, $P_{ij}$ is the onset-to-transmission between infector $i$ and infectee $j$, $I_i$ is the incubation period of infector $i$. In the first inference approach, we assumed the incubation period of the infector and the time from onset-to-transmission are independent (i.e. $Cov(P_{ij}, I_i) = 0$). Hence, the variance of the inferred generation interval is reduced to:

$$Var(G) = Var(S) \tag{7}$$

In this approach, the inferred generation interval has the same mean and variance as the observed serial interval, and is the same as the serial interval.

In the second assumption, the infectiousness profile is independent of the timing of symptoms (i.e. timing of transmission is not correlated with the timing of symptoms, $Cov(G, I_i) = 0$) and the variance of the inferred generation interval can be expressed as:

$$\begin{aligned}
Var(G) &= Var(S) + 2Cov\left(P_{ij}, I_i\right) \\
&= Var(S) + 2Cov(G - I_i, I_i) \\
&= Var(S) + 2[Cov(G, I_i) - Cov(I_i, I_i)] \\
&= Var(S) - 2Var(I) \\
&\leq Var(S)
\end{aligned} \tag{8}$$

Thus, the generation interval can be more broadly defined as the sum of the time from infection to infectiousness in the infector and the time from infectiousness to infection. The variance of the inferred generation interval is thus smaller than the observed serial interval.

Both are extreme approaches and formed the basis of inferring the upper and lower limits of the variance of the inferred generation intervals (i.e. lower and upper limits of the power to detect a difference in the inferred generation intervals) [15].

## Supporting information

**S1 Text. Supplementary figures, tables and text.**
(DOCX)

## Acknowledgments

We thank Petra Klepac and Vernon J Lee for taking time off to review and comment on this study.

## Author Contributions

**Conceptualization:** Rachael Pung, Adam J. Kucharski.

**Investigation:** Rachael Pung, Adam J. Kucharski.

**Methodology:** Rachael Pung, Adam J. Kucharski.

**Supervision:** Adam J. Kucharski.

**Visualization:** Rachael Pung, Timothy W. Russell, Adam J. Kucharski.

**Writing – original draft:** Rachael Pung.

**Writing – review & editing:** Rachael Pung, Timothy W. Russell, Adam J. Kucharski.

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
