## [Decision Letter · Decision Letter 0]

21 Sep 2023

Dear Miss Pung,

Thank you very much for submitting your manuscript "Detecting changes in generation and serial intervals under varying pathogen biology, contact patterns and outbreak response" for consideration at PLOS Computational Biology.

As with all papers reviewed by the journal, your manuscript was reviewed by members of the editorial board and by several independent reviewers. In light of the reviews (below this email), we would like to invite the resubmission of a significantly-revised version that takes into account the reviewers' comments.

We cannot make any decision about publication until we have seen the revised manuscript and your response to the reviewers' comments. Your revised manuscript is also likely to be sent to reviewers for further evaluation.

Sincerely,

Benjamin Althouse

Academic Editor

PLOS Computational Biology

Thomas Leitner

Section Editor

PLOS Computational Biology

Reviewer's Responses to Questions

**Comments to the Authors:**

Reviewer #1: Comments:

I have read this paper in a detailed manner. I wish to state that your work on Detecting changes in generation and serial intervals under varying pathogen biology, contact patterns and outbreak response is commendable. But a lot needs to be done. Therefore, author needs to do overall corrections of grammar and spelling if this manuscript will be accepted. Also, consider the following corrections listed below:

1.This manuscript is not structure well into Chapters/ Sections the paper supposes should be structured.

2.You did not consider the existence, uniqueness and properties of the model formulated for this work.

3.Numerical stimulated is absent from this work and so on.

There are many things to be done on this manuscript but if it will be accepted, I would direct the authors to study and include the following papers for more clarification:

•Analysis and Dynamics of Fractional order Mathematical Model of COVID-19 in Nigeria using Atangana-Baleanue operator, Computers, Materials & Continua, (2020).

•A new mathematical model of COVID-19 using real data from Pakistan. Results Phys 2021; 24:104098. doi:10.1016/j.rinp.2021.104098

•Forecasting of COVID-19 Pandemic in Nigeria Using Real Statistical Data. Communication in Mathematical Biology and Neurosciences. 2021 (2021). doi:10.28919/cmbn/5144.

•Mathematical model of COVID-19 in Nigeria with optimal control, Results in Physics, 28(2021), 104598. doi: 10.1016/j.rinp.2021.104598

•A fractional-order mathematical model for malaria and COVID-19 co-infection dynamics, Healthcare Analytics, 4(2023), 100210. doi: 10.1016/j.health.2023.100210

Reviewer #2: In this article, the authors explore how generation and serial intervals depend on various factors, such as changes in infection characteristics and isolation strategies. They further calculate statistical power for estimating the differences in generation and serial intervals across a wide range of scenarios. While this study is a useful contribution, I have some concerns.

Even though the calculation of statistical power is a major component of this study, the methods for power calculation is not explained anywhere in the main text or in supplementary materials. The calculation of statistical power depends on the statistical model as well as the assumed significance level. Moreover, I'm more concerned about the reported powers for the generation intervals. In reality, generation intervals are rarely observed. Instead, they are estimated from the observed serial intervals. Propagating uncertainties in the estimation of generation intervals from serial intervals are likely to decrease the power. If the authors are calculating the power to detect differences in the mean generation interval, assuming that generation intervals are known exactly, they are likely overestimating the power.

Moreover, it seems like the effects of epidemic dynamics have been largely overlooked in this study. Typically, when a new variant invades a population, the previous variant in typically in a declining state. As the authors explain in the introduction, differences in epidemic growth rates can further translate to differences in the observed serial intervals. It would be interesting and useful to understand how these biases affect power and the observed differences.

Minor comment

- L52 "When variant prevalence grows rapidly within a population, it may be the result of increased transmissibility" It's unclear that increased transmissibility should necessarily correlate with shorter generation intervals. One could imagine a pathogen with higher R0 but identical infectiousness profile (and therefore generation interval distribution). It would be useful to distinguish generation intervals in real epidemics vs those representing infectiousness profiles

- L61 "Observed serial intervals may be shorter during the exponential phase of an outbreak because transmission events involving

infectees with longer incubation periods have yet to be observed" This is incorrect. Serial intervals may be longer during the growth phase because infectors are more likely to have shorter incubation periods (when we consider a group of infectors who developed symptoms at the same time)

- Overall, the infectiousness profile needs to be explained more clearly

- L133: "We then fitted a cubic Hermite spline at the respective time points and constrained the slope of the spline to be zero at each point to simulate the infectiousness profile over the course of the infection" This sentence needs some unpacking

- Equation 1: it's not obvious that the exp(Lambda(t-1)) term is needed or what it's trying to capture. Equation 1 would approximate lambda in the absence of the exp(Lambda(t-1)) term

- Simulating transmission: how many days were epidemics simulated for? Until the epidemic dies out naturally?

- "Corresponding power to detect these differences under varying human contact patterns and outbreak response" How was the power calculated? Is there an underlying statistical model?

**Have the authors made all data and (if applicable) computational code underlying the findings in their manuscript fully available?**

Reviewer #1: **No: **Data is not available on this manuscript

Reviewer #2: **No: **I couldn't find access to the code used for simulating epidemic or calculating the power.

PLOS authors have the option to publish the peer review history of their article (what does this mean?). If published, this will include your full peer review and any attached files.

Reviewer #1: No

Reviewer #2: No
---

## [Decision Letter · Decision Letter 1]

5 Feb 2024

Dear Miss Pung,

Thank you very much for submitting your manuscript "Detecting changes in generation and serial intervals under varying pathogen biology, contact patterns and outbreak response" for consideration at PLOS Computational Biology. As with all papers reviewed by the journal, your manuscript was reviewed by members of the editorial board and by several independent reviewers. The reviewers appreciated the attention to an important topic. Based on the reviews, we are likely to accept this manuscript for publication, providing that you modify the manuscript according to the review recommendations.

Please feel free to ignore the requests for the citations from Reviewer 1. I am satisfied with the authors's previous response to the reviewer and agree that the citations are not relevant to the current paper.

Please do address the concerns about power calculations from Reviewer 2.

Sincerely,

Benjamin Althouse

Academic Editor

PLOS Computational Biology

Thomas Leitner

Section Editor

PLOS Computational Biology

Please feel free to ignore the requests for the citations from Reviewer 1. I am satisfied with the authors's previous response to the reviewer and agree that the citations are not relevant to the current paper.

Please do address the concerns about power calculations from Reviewer 2.

Reviewer's Responses to Questions

**Comments to the Authors:**

Reviewer #1: Authors need to have broad knowledge about the other areas of mathematical modelling especially while researching on disease outbreak.

Moreover, The selected papers contribute to the introduction by providing diverse modeling approaches, global perspectives, forecasting accuracy, insights into optimal control strategies, and an interdisciplinary dimension addressing co-infection dynamics. This enriches the theoretical foundation of the research, aligning it with a broad spectrum of mathematical modeling applications and diverse aspects of the ongoing scientific discourse on infectious disease dynamics.

Therefore, I recommend that those papers eariler to be studied and included under the introduction of this manuscript. The following papers are hereby listed below

• Analysis and Dynamics of Fractional order Mathematical Model of COVID-19 in Nigeria using Atangana-Baleanue operator, Computers, Materials & Continua, (2020).

• A new mathematical model of COVID-19 using real data from Pakistan. Results Phys 2021; 24:104098. doi:10.1016/j.rinp.2021.104098.

• Forecasting of COVID-19 Pandemic in Nigeria Using Real Statistical Data. Communication in Mathematical Biology and Neurosciences. 2021 (2021). doi:10.28919/cmbn/5144.

• Mathematical model of COVID-19 in Nigeria with optimal control, Results in Physics, 28(2021), 104598. doi: 10.1016/j.rinp.2021.104598.

• A fractional-order mathematical model for malaria and COVID-19 co-infection dynamics, Healthcare Analytics, 4(2023), 100210. doi: 10.1016/j.health.2023.100210

Reviewer #2: The authors have addressed all my previous concerns. My only remaining concern is about the power calculations for the inferred distributions. Specifically, the authors state that they estimate the mean and variance of the generation interval distribution from the serial interval distribution and calculate power based on a t test. However, it doesn't seem like the authors account for the uncertainties associated with inference, which would overestimate the power. I don't necessarily suggest that the authors perform a new set of analyses but this seems like an important limitation of the current approach.

**Have the authors made all data and (if applicable) computational code underlying the findings in their manuscript fully available?**

Reviewer #1: None

Reviewer #2: Yes

PLOS authors have the option to publish the peer review history of their article (what does this mean?). If published, this will include your full peer review and any attached files.

Reviewer #1: No

Reviewer #2: No

Figure Files:

Data Requirements:

Reproducibility:

References:

---

## [Editor Report · Decision Letter 2]

4 Mar 2024

Dear Miss Pung,

We are pleased to inform you that your manuscript 'Detecting changes in generation and serial intervals under varying pathogen biology, contact patterns and outbreak response' has been provisionally accepted for publication in PLOS Computational Biology.

Best regards,

Benjamin Althouse

Academic Editor

PLOS Computational Biology

Thomas Leitner

Section Editor

PLOS Computational Biology

---

## [Editor Report · Acceptance letter]

19 Mar 2024

PCOMPBIOL-D-23-01002R2 

Detecting changes in generation and serial intervals under varying pathogen biology, contact patterns and outbreak response

Dear Dr Pung,

I am pleased to inform you that your manuscript has been formally accepted for publication in PLOS Computational Biology. Your manuscript is now with our production department and you will be notified of the publication date in due course.

With kind regards,

Lilla Horvath
